# The Impact of Silver Nanoparticle-Induced Photothermal Therapy and Its Augmentation of Hyperthermia on Breast Cancer Cells Harboring Intracellular Bacteria

**DOI:** 10.3390/pharmaceutics15102466

**Published:** 2023-10-14

**Authors:** Sijia Liu, Spencer Phillips, Scott Northrup, Nicole Levi

**Affiliations:** 1Department of Plastic and Reconstructive Surgery, Wake Forest School of Medicine, Winston-Salem, NC 27101, USA; siliu@wakehealth.edu (S.L.); sgphilli@wakehealth.edu (S.P.); snorthru@wakehealth.edu (S.N.); 2School of Biomedical Engineering and Sciences, Wake Forest/Virginia Tech, Winston-Salem, NC 24061, USA

**Keywords:** silver nanoparticles, breast cancer, intracellular bacteria, photothermal therapy, hyperthermia

## Abstract

Breast cancer can harbor intracellular bacteria, which may have an impact on metastasis and therapeutic responses. Silver nanoparticles are FDA-approved for their antimicrobial potential, plus they have pleiotropic benefits for eradicating cancer cells. In the current work we synthesized photothermal silver nanoparticles (AgNPs) with an absorption at 800 nm for heat generation when exposed to near-infrared laser irradiation. Breast cell lines MCF 10A, MCF7, and MDA MB 231 were infected with *Pseudomonas aeruginosa*, and their response to AgNPs, heat, or photothermal therapy (PTT) was evaluated. The results demonstrate that the application of a brief heating of cells treated with AgNPs offers a synergistic benefit in killing both infected and non-infected cells. Using 10 µg/mL of AgNPs plus laser stimulation induced a temperature change of 12 °C, which was sufficient for reducing non-infected breast cells by 81–94%. Infected breast cells were resistant to PTT, with only a reduction of 45–68%. In the absence of laser stimulation, 10 µg/mL of AgNPs reduced breast cell populations by 10–65% with 24 h of exposure. This concentration had no impact on the survival of planktonic bacteria with or without laser stimulation, although infected breast cells had a 42–90% reduction in intracellular bacteria. Overall, this work highlights the advantages of AgNPs for the generation of heat, and to augment the benefits of heat, in breast cancer cells harboring intracellular infection.

## 1. Introduction

According to the American Cancer Society, breast cancer is the second leading cause of cancer death in women in the United States [1]. Previous studies have identified the presence of intracellular bacteria within breast cancer cells [2,3,4], which have a negative impact on therapeutic responses. A number of bacterial species have been associated with breast tumors [5,6,7,8,9,10], and intracellular bacteria have been found to enhance the survival of circulating tumor cells and promote metastasis. *Pseudomonas* has been identified in higher abundance in breast tumor tissue compared to normal breast tissue [11]. Although the exact species has not been identified, given the clinical incidence of *Pseudomonas aeruginosa*, this is a suitable culprit for the investigation of intracellular infection and response to therapies. For example, *Pseudomonas* can grow in the presence of chemotherapy, and doxorubicin stimulates *Pseudomonas* growth [11]. There is limited information on the treatment of breast cancer harboring intracellular bacteria, and few antibiotics work intracellularly, plus extended use of antibiotics can result in antibiotic resistance; therefore, there is a need to evaluate therapeutic options that can simultaneously eliminate breast cancer and bacteria.

Conventional therapies for breast cancer include surgery, radiation therapy, and chemotherapy. However, adverse side effects and therapeutic resistance limit their efficacy [12]. In recent years, photothermal therapy (PTT) has emerged as a promising strategy for the treatment of breast cancer [13,14,15,16]. By irradiating photothermal agents with deep-tissue penetrating near-infrared (NIR) light, photothermic agents can generate heat to kill cancer cells of the primary tumor, and mitigate metastasis [16,17,18,19]. Moreover, PTT has minimal side effects, such as low toxicity to healthy cells and short treatment times [20]. Gold nanoparticles of various shapes and sizes have routinely been explored as PTT agents [14,21,22,23,24,25,26,27], given their inert chemical nature in the body, but they have limited antimicrobial potential in the absence of light or modification [28,29,30].

Silver nanoparticles (AgNPs) are advantageous for breast cancer, exhibiting anti-proliferative effects on cancer cells and preventing metastasis [31,32,33,34,35,36,37]. In addition, AgNPs are clinically utilized as FDA-approved antibacterial agents [38,39,40,41]. Thus, AgNPs are promising for killing breast cancer cells harboring intracellular bacteria [39]. Most AgNPs used in medicine are spherical, with an absorption near 400 nm [42,43,44,45], although the optimal wavelengths for PTT are in the infrared, where body tissue is most transparent [46]. Previously, our team developed triangular AgNPs with near-infrared absorption, which induced heat generation upon exposure to near-infrared radiation [44], and demonstrated that the cellular uptake of AgNPs is not needed for beneficial effects, so long as the nanoparticles are in close proximity to the cells. In the current study, we infected three breast cell lines to investigate whether AgNPs could induce sufficient heat to destroy both non-infected and infected breast cancer cells upon exposure to laser stimulation. One of the challenges of PTT is the limited penetration depth of externally applied light [47,48,49]. An alternative to PTT stems from the knowledge that hyperthermia augments chemo- and radio-therapies [50,51,52,53,54]. AgNPs are thermally conductive, hence, we also explored their use as pleiotropic agents to augment hyperthermia (HT), kill cancer and bacteria cells directly, or serve as PTT agents. Together these data indicate the benefits of AgNPs to eliminate breast cancer and bacteria in the breast tumor microenvironment.

## 2. Materials and Methods

### 2.1. Synthesis and Characterization of AgNPs

Trisodium citrate (TSC), low molecular weight chitosan, and polyvinylpyrrolidone (PVP) were purchased from Sigma Aldrich. Ascorbic acid, sodium borohydride (NaBH_4_), and acetic acid were purchased from Fisher Scientific (Rockingham County, NH, USA). Silver nitrate (AgNO_3_) was purchased from EMD Chemical. Solutions were prepared using ultrapure de-ionized (DI) water. Prior to synthesis, all glassware was washed with aqua regia solution (3 parts HCl to 1 part HNO_3_, and triple-rinsed with DI water).

AgNPs were prepared via a seed-mediated growth method following a reported procedure [44]. Silver seed was produced by adding 1 mL of 30 mM TSC, 2 mL of 5 mM AgNO_3_, and 1 mL of 100 mM NaBH_4_ (aged in the dark for 4 h before use) to 95 mL of rapidly stirred DI water. After 60 s of vigorous stirring, 1 mL of 10 mg/mL PVP was added, and the solution was stirred for an additional 30 min. The silver seed solution was stored at 4 °C in the dark until used in further synthesis.

To prepare nanoparticles using the seed, 20 mL of chitosan (1% acetic acid, 2 mg/mL low molecular weight chitosan) was vigorously stirred to which 133 mL of 116.4 mM TSC, 33 mL of 300 mM ascorbic acid, and 400 mL of silver seed were added. Then, 200 mL of 30 mM AgNO_3_ was added dropwise to the solution, and the solution was stirred for approximately 3–7 min at room temperature while observing a color change from clear to yellow, orange, red, purple and blue. After that, 300 mL of 10 M NaOH was added to halt the reaction. The solution was centrifuged at 14,000 rpm for 10 min to isolate the nanoparticles, which were resuspended in DI water and stored at 4 °C in the dark.

The absorbance of AgNPs was measured using a Mettler Toledo UV5Nano UV/visible scanning spectrophotometer. The hydrodynamic diameter and zeta potential were determined using a Malvern Instruments Zetasizer. The shape of the AgNPs was observed using a FEI Technai BioTwin transmission electron microscope.

### 2.2. Hyperthermic Potential of AgNPs

The 800 nm light source for NIR stimulation was a K-Cube^®^ laser (Summus Medical Laser Inc., Franklin, TN, USA), with a 1 cm beam diameter, operating under continuous wave (CW) mode and 5 W power (6.37 W/cm^2^). The beam diameter completely covered a single well of a 96-well plate. To determine the heat generation, 200 µL of 0, 10, 25, or 50 µg/mL of AgNPs in water were placed into wells of a 96-well plate. Laser irradiation was applied to each well for 10, 20, 30, 40, 50, or 60 s. The initial and final temperatures were measured using a fiberoptic probe (Qualitrol Neoptix^®^ (Fairport, NY, USA) and Nomad thermometer), and the temperature differences were plotted to generate a temperature increase curve over time. 

### 2.3. Cell Culture and Cell Infection

Epithelial breast cells MCF 10A, and breast cancer cells MCF7 and MDA MB 231 were purchased from American Type Culture Collection (ATCC). MCF 10A cells were cultured in DMEM/F12 supplemented with 1% L-glutamine, 1% penicillin/streptomycin, 5% heat-inactivated horse serum, 20 ng/mL epidermal growth factor, 10 mg/mL insulin, 0.5 mg/mL hydrocortisone, 100 ng/mL cholera toxin, and 0.2 mg/mL gentamicin sulfate salt. MCF7 cells were cultured in EMEM supplemented with 1% L-glutamine, 1% penicillin/streptomycin, 10% fetal bovine serum, 10 mg/mL insulin, and 0.2 mg/mL gentamicin sulfate salt. MDA MB 231 cells were cultured in DMEM/F12 supplemented with 1% L-glutamine, 1% penicillin/streptomycin, 10% fetal bovine serum, and 0.2 mg/mL gentamicin sulfate salt. All cell lines were maintained at 37 °C in a 5% CO_2_ humidified environment except during hyperthermia or photothermal treatments.

Based on our previously published histological results [11] demonstrating the presence of *Pseudomonas* in breast cancer cells from human patients, we utilized *P. aeruginosa* in a highly reproducible model of intracellular infection. *Pseudomonas aeruginosa* (ATCC 27853) was purchased from ATCC. Difco™ Luria–Bertani (LB) broth was purchased from Fisher Scientific. To infect the breast cells each of the cell lines was plated at a density of 200,000 cells in the media described above without penicillin/streptomycin or gentamicin sulfate (antibiotic free infection media) in two T25 culture flasks and incubated at 37 °C with 5% CO_2_ overnight. One T25 flask of cells was trypsinized to count the number of breast cells, while the other flask was used for infection of the cells and experimental evaluation. *P. aeruginosa* was grown in LB broth overnight at 37 °C with shaking at approximately 160 rpm to facilitate aeration. The bacteria pellet was collected via centrifugation, resuspended in phosphate-buffered saline (PBS), and standardized using an optical density (OD) of 0.1 at 600 nm. The multiplicity of infection (MOI) of bacteria was 10:1, and bacteria were added to the cells in media without antibiotics and incubated at 37 °C for 2 h. Then, the bacteria-containing media was removed and the cells were washed twice with PBS. Media containing 0.2 mg/mL of gentamicin sulfate was added to kill any residual extracellular bacteria since gentamicin does not penetrate the eukaryotic cell membrane. After 1 h of incubation, the gentamicin-containing media was removed and cells were washed twice with PBS. Then, infected cells were cultured in their respective media containing penicillin/streptomycin and gentamicin sulfate, with the non-infected cells being cultured in the same media, although in a different room and incubator to minimize cross-contamination. All infected cell lines were maintained at 37 °C with 5% CO_2_. Every week, bacterial colony-forming units (CFUs) were enumerated from the infected cells to quantify the extent of infection by lysing the breast cells for 45 min in water, serially diluting the lysate, plating onto LB agar plates, and counting the number of visible colonies present on agar plate the following day.

### 2.4. Cytotoxicity of AgNPs to Breast Cell Lines and P. aeruginosa

To evaluate the cytotoxicity of the AgNPs, each of the non-infected and infected cell lines was plated at a cell density of 60,000 or 200,000 cells per well in triplicate in 6-well plates. The cells were allowed to adhere for 24 h prior to incubation with AgNPs. Then, media was removed and the cells were incubated in fresh media containing 0, 10, 25, 50, 100, or 250 µg/mL of AgNPs at 37 °C for 2 or 24 h. Clinical hyperthermia treatments often use 30–120 min of elevated temperature. Even though the PTT treatment is rapid, since there are multiple plates of cells, we sought to standardize the timing of the AgNP exposure to 2 h, since this is the maximum time that cells would be exposed while also being exposed to brief laser irradiation. Cytotoxic agents are commonly evaluated following a 24 h period, or longer, of exposure. Therefore, it is prudent to evaluate cells’ response to AgNPs alone, when PTT is not considered, since this time point will allow for uptake of the AgNPs and hence a potential modified cytotoxic response. The media containing AgNPs was removed and the cells were washed with PBS twice to remove extracellular AgNPs, incubated in fresh media without AgNPs for 24 h at 37 °C, then trypsinized and counted using a hemocytometer. 

To evaluate AgNPs’ toxicity to planktonic bacteria a single colony of *P. aeruginosa* was cultured in LB broth overnight at 37 °C and the suspended bacteria were standardized to an OD 0.1 at 600 nm. Then, 500 µL of the bacteria/ AgNPs (0, 10, 25, 50, 100 or 250 µg/mL) suspensions was added to 1 mL microcentrifuge tubes and placed on a tube revolver at 37 °C for either 2 h or 24 h, to measure the acute and prolonged exposure to AgNPs. At these timepoints 10 µL volumes were obtained, serially diluted, and plated on LB agar plates incubated at 37 °C for CFU enumeration the following day.

### 2.5. AgNPs to Augment Hyperthermia 

As determined by the heat generation of AgNPs with 5W and 800 nm exposure, we sought to evaluate the response of the breast cells to the attained temperatures for 36 s (the same as the laser duration) in the presence of AgNPs. Starting with a baseline temperature of 37 °C, the laser alone (no AgNPs) attained 43 °C, and 10, 25, and 50 µg/mL of AgNPs attained 49, 56, and 68 °C, respectively. Each of the non-infected and infected cell lines were plated at a cell density of 250,000 cells per flask in eight T25 flasks and allowed to adhere for 24 h. Five T25 flasks cells were incubated with 4 mL of media without AgNPs and subjected to 37, 43, 49, 56, or 68 °C in a circulating water bath for 36 s. The remaining three T25 flasks were incubated with 4 mL of fresh media containing 10, 25, or 50 µg/mL of AgNPs for 2 h during which time cells in 10 µg/mL were exposed to 49 °C for 36 s, cells in 25 µg/mL were exposed to 56 °C for 36 s, and cells in 50 µg/mL were exposed to 68 °C for 36 s. After 2 h incubation, the media containing AgNPs was removed, each flask was washed twice with PBS, and the cells were incubated in media alone for 24 h at 37 °C, followed by trypsinization and counting using a hemocytometer. 

Similarly, cell response to the lowest AgNPs’ concentration (10 µg/mL, which is the lowest concentration needed to generate sufficient ablative temperatures with laser exposure) was evaluated at 37, 43, 49, 56, or 68 °C in a circulating water bath for 36 s. Each of the non-infected and infected cell lines was plated at a cell density of 250,000 cells per flask in five T25 flasks and allowed to adhere for 24 h. Then, media was removed and cells were incubated in 4 mL of fresh media containing 10 µg/mL of AgNPs for 2 h. During this incubation, one flask was maintained at 37 °C as a control group, and the other four flasks were treated at 43, 49, 56, or 68 °C by submersion in a circulating water bath for 36 s. After the 2 h incubation, the media was removed and each flask was washed twice with PBS. Cells were incubated in fresh media without AgNPs for 24 h at 37 °C, then were trypsinized and counted using a hemocytometer. 

### 2.6. Photothermal Treatment of Breast Cells or P. aeruginosa

To measure the acute cellular response to photothermal therapy, each of the non-infected and infected cell lines was plated at a cell density of 5000 cells per well in 96-well plates and given 24 h to adhere. The media was removed and 200 µL of fresh media containing 0, 10, 25, or 50 µg/mL of AgNPs was added, and the plates were incubated for 2 h at 37 °C. Each well was exposed to 5 W of a 800 nm laser for 36 s once during the 2 h incubation. Following PTT the media containing AgNPs was removed, cells were washed twice with PBS, then incubated in media without AgNPs for 24 h. Twelve replicate wells per treatment group were used to have sufficient numbers for cell counting, thus these wells were trypsinized, combined, and counted using a hemocytometer. Cell viability was normalized to cells incubated without laser exposure or AgNPs. 

Clonogenics assay was used to evaluate the long-term cell survivability following PTT by examining the number of colonies formed after AgNP-induced PTT. Non-infected and infected cell lines were plated at a density of 5000 cells per well in 96-well plates (24 wells for each concentration of AgNPs) and given 24 h to adhere. The media was removed and each well was incubated with 200 µL of media containing 0, 10, or 25 µg/mL of AgNPs. The plates were incubated for 2 h at 37 °C during which twelve wells were exposed to 5 W of 800 nm light for 36 s, while the remaining 12 wells were not exposed to laser light and served as controls for cells treated with AgNPs only, and not PTT. Following the 2 h incubation the media containing AgNPs was removed, cells were washed twice with PBS, trypsinized, and wells from the same treatment condition were combined, counted and seeded in triplicate at densities of 100, 500, 1000, or 5000 cells/well on 6-well plates. The cells were then cultured in fresh media for 7–14 days to allow colony formation, after which the media was removed, cells were washed twice with PBS, fixed with cold methanol, and stained with crystal violet. Breast cell colonies (>50 cells) were counted using a KEYENCE BZ-X810 Microscope. After seeding cells at certain densities on 6-well plates, all remaining infected cells were transferred into T25 flasks and allowed to expand until there was sufficient number to enumerate bacteria. Cells were trypsinized, counted, lysed, the lysate serially diluted and plated onto LB agar plates, and the number of visible colonies were counted the following day. 

The response of *P. aeruginosa* to AgNP-induced PTT was evaluated using planktonic bacteria, not contained in breast cells, in an effort to understand the impact of PTT on the bacteria alone, knowing that thermal inactivation of *P. aeruginosa* is typically achieved at 60 °C [55]. A single colony of *P. aeruginosa* was cultured in LB broth overnight at 37 °C and the suspended bacteria were standardized to create treatment suspensions containing 0, 10, 25, or 50 µg/mL of AgNPs with OD 0.1 of bacteria. Then, 200 µL volumes of these solutions were added to wells of a 96-well plate, incubated at 37 °C before laser exposure, and exposed to 800 nm light at 5 W for 36 s. Immediately following PTT, 10 µL volumes were serially diluted and plated on LB agar for CFU enumeration the next day. 

### 2.7. Statistical Analysis

All experiments were carried out in triplicate, except PTT of the breast cells, which was performed in replicates of 12, with pooling of the treated wells to harvest sufficient cells for counting using a hemocytometer. Data was normalized to control groups without AgNPs or laser exposure to obtain cell viability or surviving breast cells which formed colonies. Bacteria were grown from a single colony of *P. aeruginosa* and were standardized to OD 0.1. Bacterial CFUs/ cell were normalized to cells incubated with media alone without laser exposure or AgNPs. For the water bath hyperthermia experiments, cell viability was normalized to cells maintained in media alone at 37 °C. For cytotoxicity of the breast cells or *P. aeruginosa*, normalization was conducted with respect to cells maintained in media alone. Error bars indicate standard error of the mean. Comparison between groups was performed using one-way ANOVA to evaluate the effect of PTT, and also the effect of AgNPs plus hyperthermia delivered using a water bath. *p*-values less than 0.05 were deemed significant.

## 3. Results

### 3.1. Characterization of AgNPs

As shown in Figure 1A, the synthesized AgNPs exhibited a strong absorption at 800 nm, and the aqueous solution of AgNPs had a blue color. Dynamic light scattering revealed that the AgNPs had an average hydrodynamic diameter of 79 nm and zeta potential of +18 ± 2 mV, although there was a smaller population of nanoparticles with a size around 10 nm, most likely unreacted seed material (Figure 1B). Transmission electron microscopy imaging indicated the triangular shape of the AgNPs (Figure 1C). Under TEM the chitosan coating was visible under high magnification (not shown in the current image) as a fairly electron-translucent fibrous material, and the coating was thin. Hence, the TEM data correlates with the DLS sizing data. We have previously published that unreacted silver seed material can be observed in unpurified AgNPs, and the small peak in the DLS data correlated with the size of the seed material [44]. Purification of the AgNPs involves centrifugation and the small nanoparticles are not able to be removed except under ultracentrifugation. Chitosan was used as the stabilizing coat for the AgNPs, as it has been used previously, and because it is antimicrobial; however, since only a thin coating of chitosan has been observed on the AgNPs using our synthesis, the cytotoxicity of the AgNPs against bacteria is due to the release of silver ions [44]. 

Upon exposure to the 800 nm laser, AgNPs generated heat due to their inherent plasmon resonance [44,56,57]. The temperature differences of 200 µL of 0, 10, 25, and 50 µg/mL AgNPs exposed to 5 W of 800 nm laser irradiation for 10 to 60 s were measured to generate a temperature increase curve (Figure 2). It has previously been determined that 180 joules of energy was effective for photothermal ablation experiments [44], where longer times and lower power were used. For example, 10 µg/mL AgNPs exposed to 3 W of 800 nm for 60 s resulted in a ΔT = 8 °C. For the current work we sought to apply 180 J using a more intense laser power and shorter time, since this will minimize the time for thermal transfer when the therapy is evaluated in vivo; thus, 5 W and 36 s was applied to achieve 180 J. Laser application only, without AgNPs, only induced mild hyperthermia (37 °C + 6 °C = 43 °C). For 36 s, the temperature increases for 0, 10, 25, and 50 µg/mL of AgNPs were 6, 12, 19, and 31 °C, respectively. Considering that cells and bacteria have an initial temperature of 37 °C for all experiments, then the minimal amount of AgNPs for the ablation of cancer cells is 10 µg/mL, since 37 °C + 12 °C = 49 °C, which is above the irreversible protein denaturation temperature of 45 °C [58]. The higher laser power generates a 1.5 times higher temperature than using a lower power and a longer time, even for the same concentration of AgNPs and laser fluence (180 J). Higher concentrations of AgNPs (100 and 250 µg/mL) were also considered for PTT; however, both concentrations resulted in boiling of the water, leading to inaccurate thermal measurements. Since lower concentrations of AgNPs were effective for heat generation they were used in subsequent PTT experiments against bacteria and breast cells.

### 3.2. Cytotoxicity of AgNPs

One of the unique discoveries in this work is the difference in doubling time between infected (MCF 10Ai, MCF7i, MDA MB 231i) and uninfected cells: MCF 10A and MCF 10Ai had the same doubling time, whereas MCF7 cells took 62 h, and MCF7i took only 48 h. There was a difference in MDA MB 231 cells also, although not as profound, with non-infected cells doubling in 26 h and infected cells taking 20 h. The initial intracellular infection rates before experimentation with AgNPs were 0.008 CFUs/cell for MCF 10Ai, 0.236 for MCF7i, and 0.5 for MDA MB 231i. Given the number of breast cells that were lysed to quantify the CFUs/cell, we determined that 1 out of every 10,000 non-tumorigenic cells was infected, whereas 1 out of every 100 tumorigenic cells was infected. These results indicate differences between naïve and tumorigenic cells, and also that different cell lines harbor more or less bacteria.

We first evaluated AgNPs in the absence of laser stimulation via a 2 h incubation of the cells with Ag NPs at 37 °C. Since 10, 25, and 50 µg/mL were sufficient for generating heat with laser stimulation cells or bacteriawere exposed to these concentrations for 2 h and the cell viability, or CFUs, were determined, as shown in Figure 3. We also examined the 24 h exposure of the cells or bacteria to AgNPs and the results are shown in Figure 4. Bacterial CFUs following 2 h exposure to AgNPs is shown in Figure 5A, and their exposure for 24 h is shown in Figure 5B. Figure 3 reveals the viability measured 24 h after a 2 h exposure to various concentrations of AgNPs at 37 °C. MCF 10A and MCF7 showed a progressive decrease in cell viability with increasing concentrations of AgNPs. Cells exposed to 10 µg/mL for 2 h had minimal reductions in cell viability, except MCF7 and MCF7i, which had an approximately 20% reduction. At 10 µg/mL, MDA MB 231 had no change in cell viability. All three cell lines were sensitive to 25 µg/mL of AgNPs, resulting in a 30–50% decrease in viability. The effect was more pronounced at 50 µg/mL, with a 36–75% decrease. There were minor variations between infected and non-infected cells, except MCF7, which had infected cells demonstrating more sensitivity (20% more at 25 µg/mL, or 5% more at 50 µg/mL). Even at the highest concentration, 50 µg/mL, MDA MB 231 had greater than 50% cell viability. For MDA MB 231 cells exposed to 50 µg/mL of AgNPs for 2 h, there was an additional 10.9% decrease in the viability of non-infected cells compared to infected cells. MDA MB 231 cells were resistant to the acute AgNPs exposure, with the MDA MB 231i being most resistant, and only exhibiting a 36.3% reduction in viability. An interesting observation is that MDA MB 231i cells are consistently less sensitive to AgNPs than their non-infected cellular counterpart, and also compared to normal breast epithelial, and non-triple-negative breast cancer cells, as determined in both the acute and prolonged exposure to AgNPs. 

Figure 4 demonstrates the response of non-infected and infected breast cell lines to various concentrations of AgNPs after a 24 h exposure. The initial 2 h exposure to AgNPs was carried out at lower concentrations, which are optimal for PTT. Exposure to toxins for 24 h is a more robust approach, and we also evaluated the use of higher concentrations that might be relevant for killing intracellular bacteria in the absence of PTT. All cell lines had decreased cell viability with increasing concentrations of AgNPs. For MCF 10A, at 50 µg/mL and above, non-infected MCF 10A cells were more resistant to AgNPs than MCF 10Ai. The response of non-infected MCF7 and MCF7i were similar. MCF7 cells were the most sensitive to AgNPs at high concentrations, with nearly complete death at 250 µg/mL. MDA MB 231 was the most resistant to AgNPs, and MDA MB 231i was more resistant than non-infected cells. Even at the highest concentration, 250 µg/mL, and after 24 h of exposure to AgNPs, MDA MB 231i had 50% cell viability and was 2.5 times more resistant than non-infected cells. Even the low dose 10 µg/mL AgNPs generated a 38–65% reduction in viability for MCF 10A and MCF7 cells, whereas MDA MB 231 cells only had a 10–25% decrease. At lower concentrations there was no difference between infected and non-infected MCF 10A and MCF7 cells, although at higher concentrations the infected cells were more sensitive to AgNPs. At 50 µg/mL MCF 10 Ai cells were almost doubly susceptible to AgNPs. At 100 µg/mL both MCF 10Ai and MCF7i cells were four and two times more sensitive to AgNPs compared to non-infected cells, respectively. At 250 µg/mL the MCF 10Ai cells were 15 times more sensitive, while MCF7, infected or not, was completely eradicated. 

Figure 5A shows the toxic effects of AgNPs on bacteria following a 2 h exposure. Planktonic *P. aeruginosa* (not intracellular) experience minimal reductions in viability, with a maximum of about 1.5 log reduction. Interestingly, there was no difference between bacteria treated with 10, 25, or 50 µg/mL AgNPs, indicating that additional AgNPs did not increase bacterial reduction. Prolonged exposure of *P. aeruginosa* to AgNPs for 24 h leads to significant 5 to 8 log reductions, as shown in Figure 5B. High concentrations of AgNPs and a prolonged time of exposure are needed to effectively eliminate planktonic bacteria.

### 3.3. Cell Sensitivity to AgNPs with Hyperthermia

A unique feature of metal nanoparticles is their thermal conductivity, allowing them to facilitate heat transfer to cells even in the absence of laser stimulation. Forty-three degrees is the equivalent temperature increase of the laser alone without AgNPs. Temperatures of 49, 56, and 68 °C were obtained for AgNPs of 10, 25, and 50 µg/mL exposed to 36 s of 5 W, 800 nm light. To evaluate the sensitivity of cells to brief heating in the presence of AgNPs (no laser stimulation), infected or non-infected cells were exposed to 10 µg/mL AgNPs and heated for 36 s in a water bath (the same time of laser application needed to reach ablative temperatures). Cellular responses measured 24 h after 2 h of exposure to 10 µg/mL of AgNPs and 36 s of treatment with various temperatures are shown in Figure 6. In general, breast cells exhibited progressive reductions in cell viability with increasing temperature. Between 49–68 °C MCF 10Ai had a minimal reduction in viability, indicating that MCF 10Ai may be more resistant to hyperthermia. There were statistical differences between infected and non-infected groups at elevated temperatures with 10 µg/mL of AgNPs. Although statistically significant, there were only minor differences between MCF7 and MCF7i cells at 49, 56, and 68 °C with the addition of AgNPs. MCF7 cells are more sensitive to brief (36 s) exposure to 43 °C (Figure 6), with a 15–23% reduction, but looking back at Figure 3 this effect is indicative of their sensitivity to AgNPs alone, not heat. Non-infected MDA MB 231 cells were not susceptible, although the infected cells were, with a 22% decrease in cell viability with 43 °C and 10 µg/mL of AgNPs, which is substantially different (22% greater reduction) than treatment at 37 °C, which only had a 1.4% decrease. Examining the data on elevated temperatures and no AgNPs (from Figure 7) for cells exposed to 43 °C for 36 s, there were no differences for MCF 10A or MDA MB 231, but there was an 11% increased reduction in cell viability for MCF7 cells with 10 µg/mL AgNPs. MCF7i cells had a 6.5% increase in viability, which is not significant. Comparing brief exposure to elevated temperatures (36 s) with or without exposure to AgNPs for 2 h, there were some differences, as identified in Table 1, which shows the viability comparisons. The effect of AgNPs is variable for MCF 10A and MCF7, whether infected or not, whereas MDA MB 231 cells, infected or not, showed decreased viability when AgNPs were added to hyperthermic temperatures. Recalling from Figure 3 that MDA MB 231i cells were most resistant to AgNPs, they are more sensitive to the effects of AgNPs at elevated temperatures. The results of Table 1 indicate that AgNPs at 10 µg/mL augment hyperthermia at lower temperatures for MCF 10A, MCF 10Ai, MCF7, and MCF7i. At 68 °C though, there is no difference, or an increase in viability, for these cells, which indicates an unexpected protective effect of the nanoparticles. MDA MB 231, infected or not, had decreases in cell viability for all temperatures with the addition of 10 µg/mL AgNPs.

As shown in Figure 2, the temperature increases of 0, 10, 25, and 50 µg/mL AgNPs with 5 W of 800 nm laser irradiation for 36 s were 6 °C, 12 °C, 19 °C, and 31 °C, respectively. Accordingly, the final temperatures in the photothermal ablation experiments were 43 °C (37 °C + 6 °C), 49 °C, 56 °C, and 68 °C. To evaluate cell response to hyperthermia with AgNPs only (without laser stimulation) each of the cell lines incubated with 0, 10, 25, or 50 µg/mL AgNPs for 2 h was exposed to temperatures of 49, 56, or 68 °C for 36 s using a water bath. The cell viability measured 24 h after hyperthermia is shown in Figure 7. As expected, breast cells showed a progressive decrease in viability with increasing temperature, and cells incubated with AgNPs exhibited greater reductions in viability. Especially for MCF 10Ai (Figure 7A), non-infected MCF7 (Figure 7B), and non-infected MDA MB 231 (Figure 7C), AgNPs led to significant decreases in cell viability. From Figure 7, without AgNPs, in MCF 10A cells treated with 36 s of 49 °C there was an approximate 20% reduction in cell viability, and MCF7 had a 26% greater reduction with the addition of just 49 °C for 36 s, regardless of whether the cells were infected or not. However, MDA MB 231 cells, infected or not, had no reduction in viability. Comparing the data of Figure 3 and Figure 7A for 10 µg/mL, a brief exposure of 36 s to 49 °C resulted in a drastic reduction in viability, where MCF 10A had a 33.3% additional reduction, and MCF 10Ai cells had an additional 47% reduction compared to cells just treated with AgNps at 37 °C. MCF7 cells had a 37.9% reduction and MCF7i cells had a 19.1% additional reduction when treated with 49 °C for 36 s plus 10 µg/mL of AgNPs compared to treatment with nanoparticles only. MDA MB 231 cells had a 24.7% additional reduction and MDA MB 231i cells had a 20.8% additional reduction. MCF 10Ai cells were most sensitive to both AgNPs and 49 °C. Although brief 49 °C exposure and AgNPs were beneficial together for non-infected cells, the MCF7i and MDA MB 231i cells were less susceptible than their non-infected counterparts. From Figure 7, brief 56 °C for 36 s reduced the viability of all the cell types by 28–55%, with the exception of MDA MB 231i cells which had no significant change in viability. MCF 10A cells had a further reduction of 4.1%, and MCF 10Ai cells had a further reduction of 30.8% with the addition of 56 °C for 36 s to 25 µg/mL of AgNPs (comparing Figure 3 and Figure 7). MCF7 cells had a further reduction of 47.9%, and MCF7i cells had a further reduction of 6.5% with the addition of 56 °C to 25 µg/mL of AgNPs. MDA MB 231 cells had a further reduction of 10.4%, and MDA MB 231i cells had a further reduction of 10.9% with the addition of 56 °C plus 25 µg/mL of AgNPs. Brief exposure to 36 s at 68 °C reduced all cells’ viability, with 70–80% for MCF 10A and 65–73% for MCF7, whether infected or not, but only 51% for MDA MB 231 and 58.5% for MDA MB 231i cells. There is a benefit to adding 68 °C to 50 µg/mL of AgNPs; it resulted in a 6.4–22% additional reduction for MCF 10A and MCF7 cells, infected or not. There is a greater benefit for MDA MB 231, and their infected cells, which had additional 42.6 and 45.1% reductions when 68 °C is combined with 50 µg/mL of AgNPs. 

There are interesting observations of reduced viability between infected and non-infected cells with hyperthermia augmented by AgNPs. In Figure 7A, MCF 10A cells have 17.7, 22.9, and 0.9% additional reductions in viability and MCF 10Ai cells have 32.6, 30.6, and 12% additional reductions in viability when AgNPs at 10, 25, and 50 µg/mL are added to 49, 56, or 68 °C for 36 s. There was a 12.5% further reduction in MCF 10Ai cells treated with 56 °C and 25 µg/mL compared to MCF 10A cells, but there was only a 7.8% reduction between infected and non-infected cells when 50 µg/mL was added to 68 °C. The major benefit is that there is no statistical difference in non-infected MCF 10A cells and infected cells with hyperthermia alone, but AgNPs resulted in an additional 16.8–20.2% reduction in cell viability. This result demonstrates that AgNPs augment hyperthermia to a greater extent in the infected non-tumorigenic cells. There was no difference between MCF7 and MCF7i cells when treated with 49 °C for 36 s in the absence of AgNPs. However, there was a more profound reduction in MCF7i cells compared to non-infected cells when treated with 36 s of 56 and 68 °C, resulting in 17.1 and 8.1% increased reductions with AgNPs. As shown in Figure 7B, there were bigger reductions in non-infected MCF7 cells (34.9, 47.7, and 26.8% reductions) when 25 or 50 µg/mL of AgNPs were added to 36 s of 49, 56, and 68 °C, compared to infected cells (13.7, 9.5, and 8.2%). AgNPs seem to protect infected cells from thermal damage in this case, compared to the non-infected cells. As shown in Figure 7C, for MDA MB 231 or MDA MB 231i cells treated with 49 °C for 36 s there was no change in viability in the absence of AgNPs. There is a reduction in viability with the addition of 10 µg/mL of AgNPs due to the combination of heat and nanoparticles, although there was no difference between non-infected and infected cells. MDA MB 231 cells treated with 56 °C for 36 s had a 17.3% decrease with the addition of 25 µg/mL of AgNPs for non-infected cells compared to the infected cells, which had a 50.7% decrease. When cells were treated with 68 °C for 36 s the non-infected MDA MB 231 cells had a 38.7% reduction and the infected cells had a 22.9% reduction with the addition of 50 µg/mL of AgNPs. Infected MDA MB 231 cells either had no reduction, or an increase in viability, when 10, 25, or 50 µg/mL of AgNPs were added to 49, 56, or 68 °C. This indicates that AgNPs may not be highly beneficial for augmenting brief hyperthermia (here using a water bath) when MDA MB 231 cells are infected. 

### 3.4. Cell Viability to Photothermal Treatment

Each of the breast cell lines incubated with 0, 10, 25, or 50 µg/mL of AgNPs for 2 h was exposed to 5 W of a 800 nm laser for 36 s to induce ablative (>45 °C) temperatures. The cellular response 24 h after photothermal treatment is presented in Figure 8. An obvious reduction in cell viability was observed at 10 µg/mL AgNPs, although infected cell lines were more resistant to AgNPs-induced PTT than non-infected cell lines, with a 22–33% enhanced viability. Nonetheless, there was a profound reduction in viability with non-infected MCF 10A, MCF7, and MDA MB231 cells having 78.2, 88.7, and 94.9% reductions with 10 µg/mL of AgNPs-induced PTT. The infected lines had 45, 66.7, and 68.1% reductions in viability for MCF 10Ai, MCF7i, and MDA MB 231i cells. PTT with AgNPs at higher concentrations resulted in nearly complete cell death for both non-infected and infected breast cells. All infected cells were more resistant to photothermal therapy with 10 µg/mL of AgNPs compared to their non-infected counterparts. There was less than 10% cell viability for all lines, infected or not, at 25 µg/mL, and no surviving cells with 50 µg/mL of AgNPs plus laser radiation. 

Figure 9 shows the response of planktonic *P. aeruginosa* to AgNP-induced PTT. There is a 0.3 log reduction with 10 µg/mL of AgNPs and laser stimulation, and a greater than 1 log reduction with 25 µg/mL and laser stimulation, although complete ablation of planktonic bacteria with 50 µg/mL and laser was observed. The result makes sense since temperatures above 65 °C are often needed to eradicate bacteria and PTT using 50 µg/mL resulted in a ΔT = 31 °C, so considering that bacteria had an initial temperature of 37 °C, a temperature of 68 °C in the bacterial solution would be attained. However, it was not expected that such a rapid application of hyperthermia for 36 s could induce a maximum temperature of 68 °C; an average of 52.5 °C over the course of laser exposure for 36 s would be effective [59]. This result may be due to the proximity of the AgNPs to free-floating bacteria in solution. 

Figure 10 shows the bacterial CFUs/cell for infected breast cells. MCF 10Ai cells had an infection of 0.0081 CFUs/cell with no treatment, MCF7i had 0.236 CFUs/cell, and MDA MB 231i had 0.5 CFUs/cell, which shows that breast cancer cells may harbor more intracellular bacteria than non-tumorigenic cells. For MCF 10A cells there was a statistically significant increase (80.2%) in CFUs/cell with laser only and no AgNPs, where only 43 °C was attained. When 10 µg/mL of AgNPs was used to generate a temperature of 49 °C there was no difference in CFUs/cell, with both having approximately 0.014 CFUs/cell. When 25 µg/mL of AgNPs was used with MCF 10Ai cells to generate a temperature of 56 °C there was a 100% reduction in CFUs/cell, even though there was a sufficient number of cancer cells to count since the cells treated with PTT were allowed to regrow following treatment to allow a sufficient number of cells for lysis and bacterial enumeration. This indicates that the cells that survive ablation with 25 µg/mL do not harbor bacteria. The same trend was observed in MDA MB 231i cells treated with 25 µg/mL of AgNPs and laser stimulation. For MCF7i cells treated with 36 s of 5 W and 800 nm light, without AgNPs, for a volumetric temperature change of 43 °C, there was a 39.4% reduction in CFUs/cell. This trend was augmented when 10 µg/mL of AgNPs plus laser stimulation for 36 s was applied, resulting in a 42.6% reduction in CFUs/cell. There was a slightly higher incidence of CFUs/cell when MCF7i cells were treated with 25 µg/mL, with only a 17.3% reduction in CFUs/cell with laser compared to without. MDA MB 231 cells had a 69.4% reduction in CFUs/cell with laser alone and no AgNPs, and 89.8 and 100% reductions in CFUs/cell with 10 and 25 µg/mL AgNPs plus 36 s of laser exposure to generate 49 and 56 °C. These results confirm that AgNP-induced PTT can reduce or eliminate intracellular bacteria in cells that proliferate following PTT. 

Breast cell clonogenic results shown in Figure 11A illustrate that without laser stimulation, non-infected and infected breast cells incubated with 0, 10, or 25 µg/mL of AgNPs alone for 2 h had only slight reductions in the number of surviving cells that could form colonies. The infected cells had more breast cell colonies than their corresponding non-infected cells which supports the hyperthermia results noted earlier. The infected MCF 10A and MCF7 cells had increased survivability, with no loss in response to exposure to AgNPs for 2 h. Although the acute study results in Figure 3 show that for MCF 10A cells, infected or not, there was an approximately 45% decrease in cell viability with 25 µg/mL, the clonogenics assay demonstrates that the MCF 10A cells recover well from AgNPs exposure, and they recover better than either cancer cell line. Non-infected MCF7 cells had the greatest reduction in survivability with 25 µg/mL of AgNPs, which correlates with the acute cytotoxicity assay data of Figure 3; nonetheless, the cells recovered well from exposure to AgNPs. The acute data indicated that MDA MB 231i cells were most resistant to AgNPs, and the clonogenics assay results demonstrates these cells have minimal struggle in recovering from AgNPs exposure.

As shown in Figure 11B, upon exposure to laser stimulation, breast cell colony numbers decreased with increasing AgNPs’ concentration (and hence increased temperature). For MCF 10A and MCF 10Ai, MDA MB 231 and MDA MB 231i, and non-infected MCF7, 25 µg/mL AgNPs with the laser induced sufficient heat to cause no colony formation (thus no datapoints). A significant reduction in colony number was also observed for MCF7i under this condition. At 10 µg/mL, infected breast cells had a greater number of surviving cells to form colonies than their corresponding non-infected cells for all three cell lines, in agreement with the data from Figure 3 and Figure 5, indicating that infected breast cells were more resistant to AgNPs-induced hyperthermia than non-infected breast cells. Infected and non-infected cells were all susceptible to PTT using 10 or 25 µg/mL of AgNPs. Clonogenics data confirm the same trend as observed in Figure 8 where cell viability was measured 24 h after PTT, with the infected cells being more resistant to PTT. There was a substantial decrease in survivability for all cells with 10 µg/mL of AgNP- induced PTT, with MDA MB 231 cells being most sensitive, and this confirms the results of Figure 8 where MDA MB 231 cells were the most sensitive to AgNP- induced PTT. The conclusion of the acute PTT results (Figure 8) and cell regrowth potential (Figure 11B) support the premise that although infected breast cells are resistant to lower doses of AgNPs for PTT, higher doses are most suitable for completely eliminating both infected and non-infected cancer cells. 

## 4. Discussion

We hypothesize that the intracellular infection stresses the cells, in ways that have yet to be elucidated, and the stress makes the cells more susceptible to both heat and AgNPs. A phenomenon that has yet to be considered is that dying cancer cells may release potentially viable bacteria into the extracellular space, hence allowing for the subsequent infection of other cells; therefore, PTT may be beneficial since it can kill bacteria during the process of ablating the cancer cells. As demonstrated in our results, cells with intracellular infection can be less susceptible to AgNPs than hyperthermia, although the combination of both, and especially when AgNPs are used for PTT, can be beneficial. The results suggest that a lower incidence of intracellular infection would benefit chemotherapeutic treatments of breast cancer. Both breast cancer cell lines harbor a higher number of CFUs per cell than the non-tumorigenic cell line. Non-tumorigenic cells had 1 out of every 10,000 cells infected, but tumorigenic cells had a higher number of infected cells infected; 1 out of every 100. The current clinical literature has indicated that bacteria survive in the intracellular environment; however, this clinical incidence has not been quantified. Further information is needed to determine the number breast cancer cells that harbor intracellular bacteria in clinical samples. Our results here demonstrate that intracellular infection can be reproduced in a laboratory setting, and higher or lower levels of infection may be possible depending upon the bacterial strain used and the host cell’s line. Further information is needed to determine why this trend occurs, and if all cancer is more susceptible to harboring intracellular bacteria and why. Antibiotics may not be effective against breast cells harboring intracellular infection since they can induce drug-resistant bacteria, and many antibiotics cannot enter eukaryotic cells to tackle intracellular pathogens; however, AgNPs are toxic to both cancer cells and bacteria. *P. aeruginosa* is an opportunistic pathobiont, and possibly an oncomicrobe, with the capacity to infect both non-tumorigenic and tumorigenic breast cells. A limitation of this study is that the exploratory nature of this work does not consider immune cell response to intracellular infection, and especially those cells that undergo thermal necrosis with the potential release of surviving bacteria, since both mechanisms would induce immunostimulatory effects. There are also other types of bacteria associated with breast cancer that can be used for intracellular infection, including *Fusobacterium nucleatum* and *Staphylococcus aureus* [10,60,61]. Our work uses *P. aeruginosa*, based on our previously published clinical findings from human breast cancer samples [11]. A key observation of our work is the hyperproliferation seen only in the cancer cells, with both MCF7i and MDA MB 231i cells having 22.6 and 23.1% faster doubling times compared to their non-infected counterparts. Our work supports the work of others, where pancreatic and colorectal cancer cells with intracellular infection demonstrated hyperproliferation [60,62]. The previously published works only evaluated hyperproliferation in cancer cells using colorimetric or DNA-based assays (MTS, MTT, CCK-8, live/dead, and resazurin staining); since the assays cannot differentiate between bacterial and cell viability the results may conflate the trend of hyperproliferation. Our study confirms that hyperproliferation occurs, as quantified by manual cell counting. Direct counting of viable cells is currently the only way to ensure that results between infected and non-infected cells can be compared. Furthermore, we demonstrated that non-tumorigenic MCF 10A cells can be infected at a lower level than in the breast cancer cell lines, with no observation of hyperproliferation, a result that has not been demonstrated before. We also had a 29.1 fold higher amount of CFUs/ cell for MCF7 and 61.7 fold higher amount of CFUs/ cell for MDA MB 231, respectively, compared to MCF 10A. Compared to Yu et al. [62], in which *E. coli* was used to infect colorectal cancer cells, when we used *P. aeruginosa* at the same MOI of 10:1, we found about the same level of infection: Yu et al. found 1 × 10^−1^ CFUs/cell, and we found 2 × 10^−1^ for MCF7 and 5 × 10^−1^ CFUs/cell for MDA MB 231. 

Considering that most of the AgNPs used extracellularly for PTT will not achieve high concentrations near intracellular bacteria, it may be concluded that AgNPs do not kill bacteria via the known mechanisms of oxidative stress [63]. There might be a benefit to adding AgNPs to hyperthermia treatments to decrease cell viability in infected cells. One limitation of the current cytotoxicity studies is in determining the amount of AgNPs that are uptaken by cells, and whether intracellular infection promotes more or less uptake of the AgNPs. Acute exposure (2 h) to low dose, 10 µg/mL, AgNPs does not cause much reduction in cell viability although higher concentrations are toxic even with an acute 2 h exposure. The higher concentrations of AgNPs given to breast cells exposed for 24 h reduce cell viability profoundly for MCF 10A and MCF7 cells, which even begins at 10 µg/mL. Notably, MDA MB 231 cells were most resistant to AgNPs, which is contradictory to previous results [35,36,44]. One of the most dramatic findings is that MDA MB 231i cells were highly resistant to AgNPs, even at high concentrations (250 µg/mL) for 24 h. Lower doses of AgNPs that exhibit toxicity to breast cells (10–50 µg/mL) also cause a minor (~1 log, thus not clinically significant) reduction in planktonic *P. aeruginosa*. Prolonged exposure to AgNPs for 24 h showed that bacteria overcome the acute insult, but there is profound toxicity at concentrations higher than 50 µg/mL (5–8 log reduction). 

Forty-three degrees Celsius is mild hyperthermia and it is known that cancer cells are more sensitive to heat than non-cancerous cells [51]. The reason why hyperthermia is often used against cancer is because cancer cells have been shown to be more sensitive to elevated temperatures due to their inability to overcome DNA strand breaks [64,65]. AgNPs augment hyperthermia whether cells are infected or not, but there is a greater benefit against infected cells. Given the toxicity of the AgNPs, the potential for thermal transfer, and knowledge of cell susceptibility to hyperthermia, we examined the addition of AgNPs to elevated temperatures. The temperatures that were achieved using PTT ranged from 49 to 68 °C, hence, we examined how cells with and without infection would respond to brief exposure to these temperatures with AgNPs. AgNPs can augment hyperthermia, and are especially beneficial at lower temperatures. Only the lowest dose of AgNPs (10 µg/mL) was needed to elicit such an effect, although higher doses of AgNPs also boost brief exposure to elevated temperatures. A major benefit is that AgNPs augmented hyperthermia especially in infected cells, further supporting the hypothesis that these cells may be stressed by intracellular bacteria and hence more responsive to therapeutics that increase stress responses, leading to cell death. 

AgNPs, synthesized into triangular shapes for absorption at 800 nm for PTT, were synthesized and were effective for heat generation. AgNPs are inherently antimicrobial, cytotoxic at higher concentrations, and can enter cells, as multiple groups have shown [35,36,44]. They can kill extracellular bacteria, but have limited potential to kill intracellular bacteria without additional stimulation such as heat-generating PTT. Even a low dose (10 µg/mL AgNPs) and 800 nm at 5 W for 36 s could generate ΔT = 12 °C, which is sufficient for overcoming the 45 °C (with 37 + 12 °C = 49 °C) threshold of irreversible protein denaturation and hence effective for killing cancer cells. Higher concentrations of AgNPs get much hotter (up to 37 + 31 °C = 68 °C). PTT can be induced rapidly, in only 36 s, using a higher laser power that will have improved penetration depth [66]. A limitation is that PTT temperatures were measured as a volumetric change (200 µL) and temperatures that the cancer or bacterial cells experience locally at their cell surface might be much higher given the proximity of the AgNPs during laser exposure. It was expected that AgNP-induced PTT would kill breast cells, but it was unexpected that all infected cells would be resistant to PTT. Resistance could be overcome with the use of 25 µg/mL of AgNP-induced PTT. PTT is more effective than water bath hyperthermia with AgNPs. PTT of planktonic bacteria can completely eliminate the bacteria at a concentration of 50 µg/mL, although there are also statistically significant reductions for 10 and 25 µg/mL. AgNPs at 10 µg/mL with laser stimulation did not reduce the number of intracellular bacteria in MCF 10A cells; however, there was a reduction in both breast cancer cell lines. Intracellular infection is reduced following PTT, but may not be completely eliminated depending upon the concentration of AgNPs used. Infected cells that survive PTT continue to harbor bacteria, a result that makes sense since higher temperatures are needed to kill the bacteria. Intracellular bacteria alter cell response to hyperthermia, and although PTT is effective, it is critical that high enough doses of AgNPs and heat are used to kill infected cells, since survivors could metastasize, and the bacteria could migrate with these cells to other locations. 

## 5. Conclusions

The results of this work demonstrate that breast cells harbor intracellular pathogens, with breast cancer cells having an increased bacterial burden. The current work does not indicate that intracellular infection promotes non-tumorigenic breast cells into a cancerous phenotype, but it suggests that both tumorigenic and non-tumorigenic breast cells respond differently to heat and AgNPs, both of which induce cell stress, when bacteria are present. In the absence of the laser, breast cells incubated with AgNPs exhibited greater reductions in cell viability than cells incubated with media alone at the same temperature, indicating the potential application of AgNPs to improve the efficiency of hyperthermia treatment. Infected breast cells were more resistant to AgNP-induced PTT, and the augmentation of hyperthermia with AgNPs. In this work, 25 mg/mL of AgNPs with laser stimulation generated sufficient heat, leading to nearly complete cell death of both non-infected and infected breast cells. The lack of breast cell colony formation at the 25 mg/mL concentration demonstrated that AgNP-induced cellular damage was irreversible, indicating less potential for cell regrowth. PTT caused reductions in bacterial colony-forming units, and also in CFUs/cell for breast cells. These results highlight that triangular AgNPs could be effective for PTT against infected breast cancer cells. The overall results of this work demonstrate the benefit of using AgNPs as cytotoxic photothermal agents to kill pathogenic bacteria both inside and outside breast cells. PTT is a relatively new option under consideration to eliminate breast cancer, especially cells that harbor bacteria which may be resistant to treatment and have higher metastatic potential. Most PTT focuses on killing either cancer cells or bacteria separately; here we demonstrate the potential for killing bacteria within cancer cells, while simultaneously killing the cancer cells. The next step is to evaluate how reducing intracellular bacteria can improve responses to breast cancer treatments, where limited options exist for tackling both bacteria and cancer cells in the breast microenvironment.

## Figures and Tables

**Figure 1 pharmaceutics-15-02466-f001:**
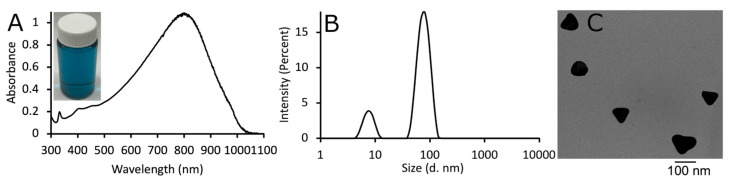
(**A**) Absorption spectrum of AgNPs in water. (**B**) Dynamic light scattering data of AgNPs in water. (**C**) Transmission electron microscopy image of AgNPs.

**Figure 2 pharmaceutics-15-02466-f002:**
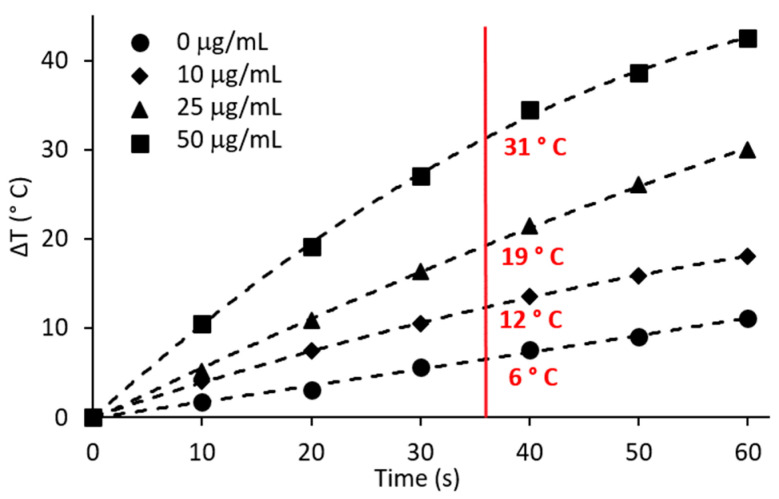
Temperature change versus laser irradiation (5 W, 800 nm) time for various concentrations of 200 µL of AgNPs in water.

**Figure 3 pharmaceutics-15-02466-f003:**
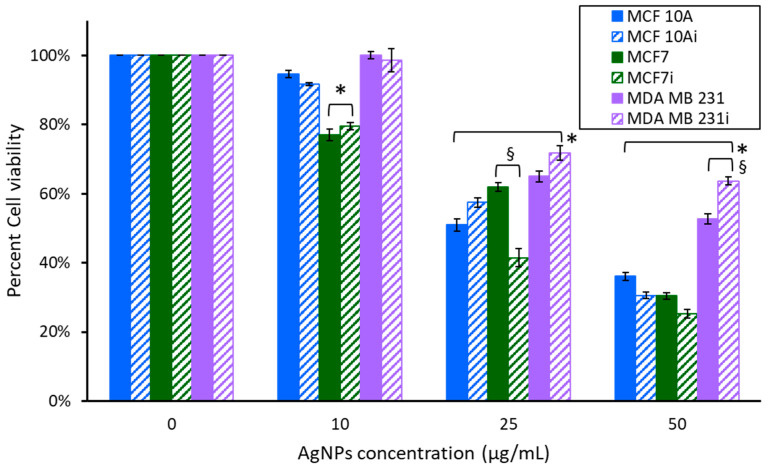
Cell viability of breast cell lines measured 24 h after 2 h of exposure to various concentrations of AgNPs. * indicates a statistical (*p* < 0.05) difference from the control group with 0 µg/mL of AgNPs, and § indicates a difference between non-infected and infected cells.

**Figure 4 pharmaceutics-15-02466-f004:**
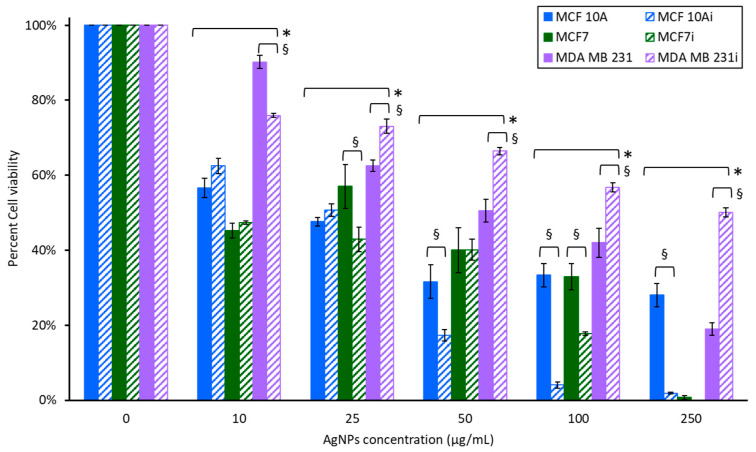
Cell viability of breast cell lines exposed to various concentrations of AgNPs for 24 h. * indicates a statistical (*p* < 0.05) difference from the control group with 0 µg/mL of AgNPs, and § indicates difference between non-infected and infected cells.

**Figure 5 pharmaceutics-15-02466-f005:**
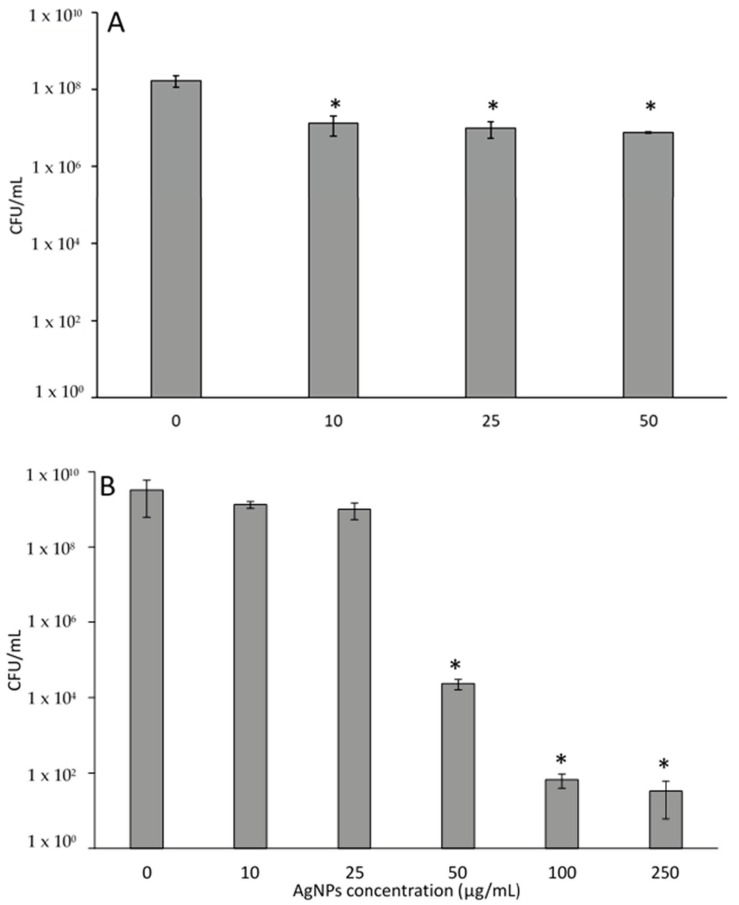
PA27853 response to AgNPs (**A**) exposed for 2 h, or (**B**) 24 h. * indicates a statistical (*p* < 0.05) difference from the control group with 0 µg/mL of AgNPs.

**Figure 6 pharmaceutics-15-02466-f006:**
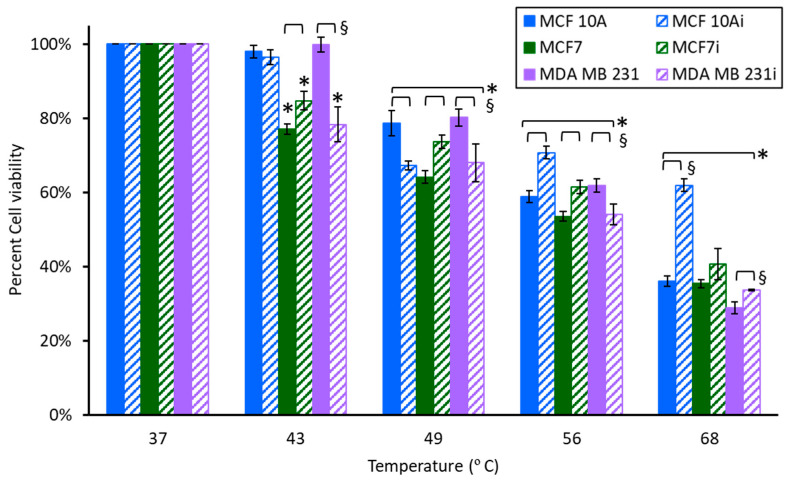
Cell viability of breast cell lines measured 24 h after 2 h of exposure to 10 µg/mL of AgNPs and 36 s of exposure to elevated temperatures (using a water bath). * indicates a statistical (*p* < 0.05) difference from the control group with 0 µg/mL of AgNPs, and § indicates difference between non-infected and infected cells.

**Figure 7 pharmaceutics-15-02466-f007:**
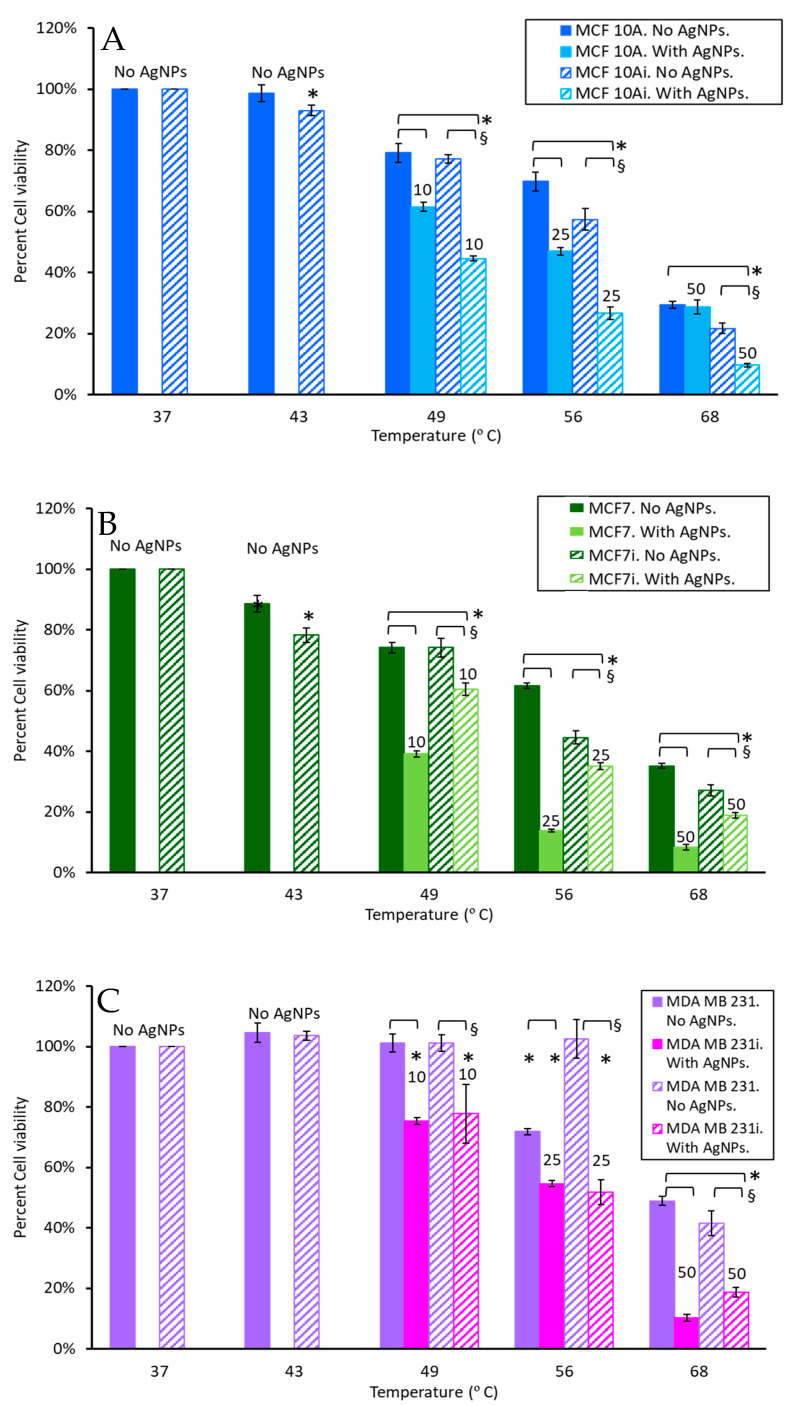
Cell viability of breast cell lines (**A**) MCF 10A, (**B**) MCF7, and (**C**) MDA MB 231 measured 24 h after a 36 s exposure to hyperthermia (using a water bath). * indicates a statistical (*p* < 0.05) difference from the control group with 0 µg/mL of AgNPs, and § indicates difference between non-infected and infected cells.

**Figure 8 pharmaceutics-15-02466-f008:**
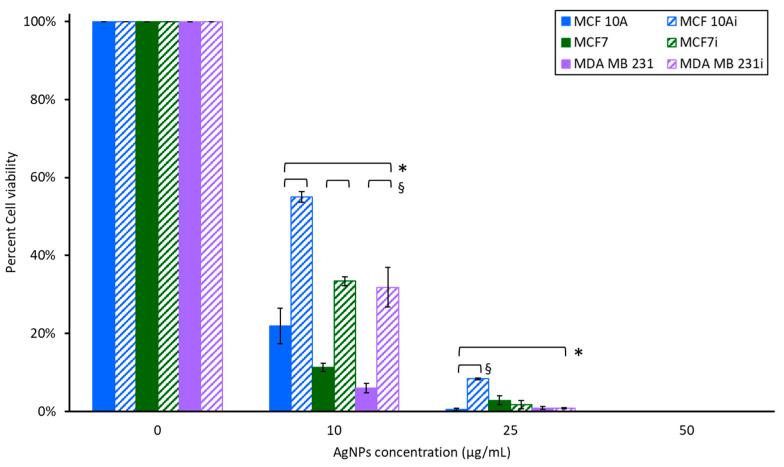
Cell viability of breast cell lines measured 24 h after photothermal treatment. * indicates a statistical (*p* < 0.05) difference from the control group with 0 µg/mL of AgNPs, and § indicates difference between non-infected and infected cells.

**Figure 9 pharmaceutics-15-02466-f009:**
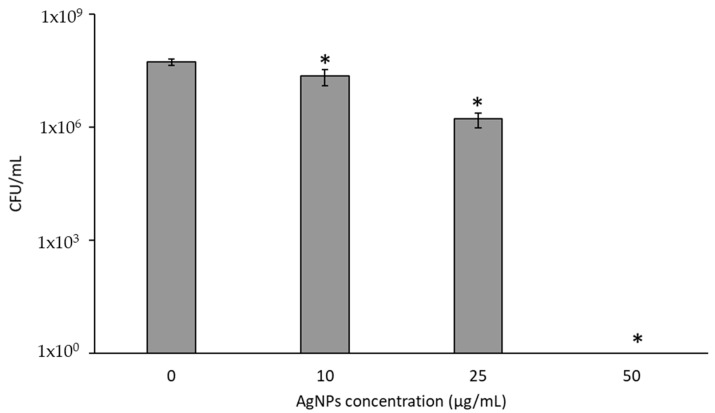
CFUs/ mL of *P. aeruginosa* following photothermal therapy. * indicates a statistical (*p* < 0.05) difference from the control group with 0 µg/mL of AgNPs.

**Figure 10 pharmaceutics-15-02466-f010:**
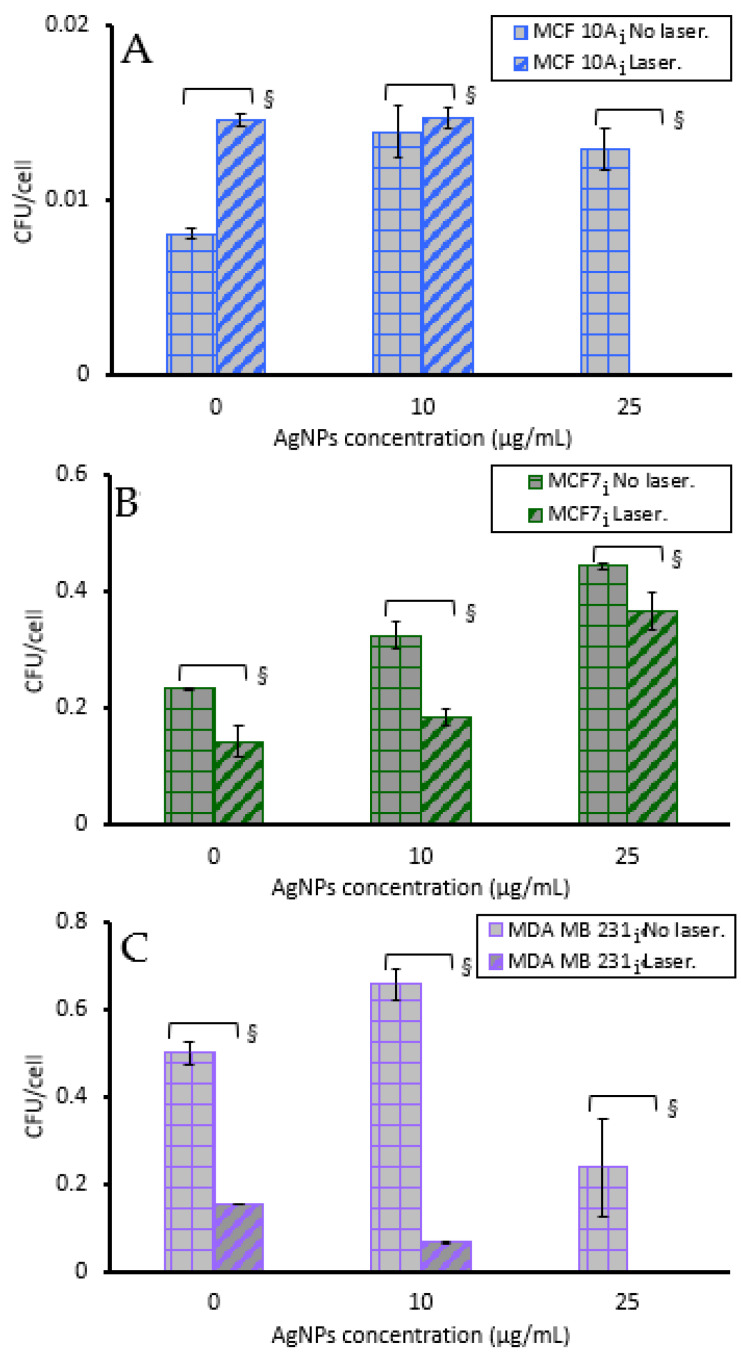
Bacteria colony-forming units per cell of infected (**A**) MCF 10A, (**B**) MCF7, and (**C**) MDA MB 231. Each of the cell line was incubated with various concentrations of AgNPs for 2 h and treated without/with 5 W of a 800 nm laser for 36 s. § indicates a statistical (*p* < 0.05) difference between laser and no laser.

**Figure 11 pharmaceutics-15-02466-f011:**
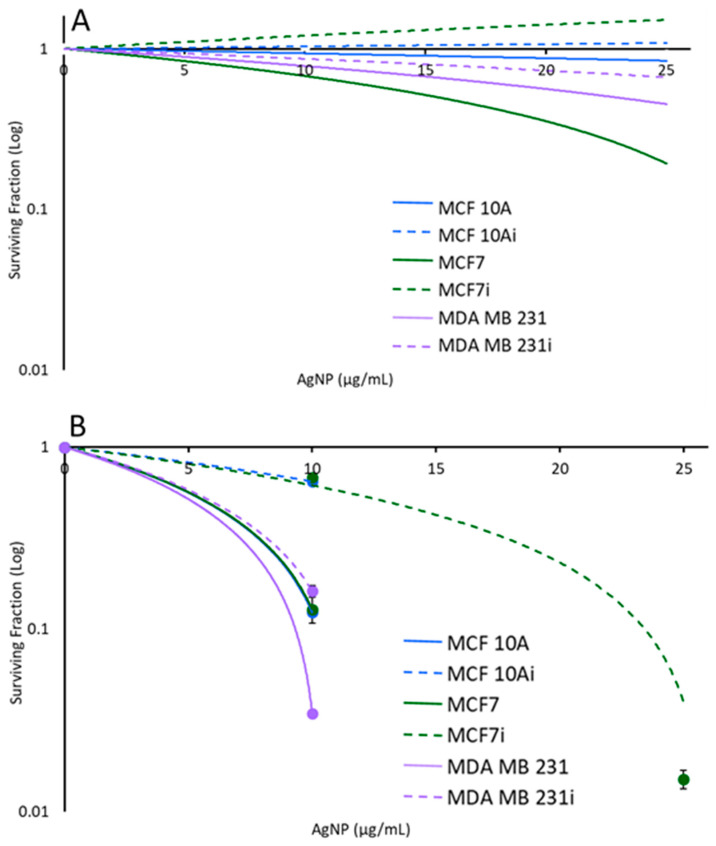
Semi-log plots of the survival fraction of breast cell lines treated with different concentrations of AgNPs. (**A**) In the absence of laser, cells incubated with AgNPs alone had relatively slight changes in the number of colonies formed. (**B**) Upon exposure to laser, colony numbers decreased significantly with increasing AgNPs’ concentration.

**Table 1 pharmaceutics-15-02466-t001:** AgNPs (10 µg/mL) can augment hyperthermia. I means increase, D means decrease, and ND means no statistical difference.

	Temp (Deg. C)	Cell Viability No AgNPs	Cell Viability with AgNPs	Increase/ Decrease
MCF 10A	49	79.1	78.6	D
	56	69.8	58.8	D
	68	29.5	36.1	I
MCF 10Ai	49	77.2	67.2	D
	56	57.3	70.7	I
	68	21.7	61.8	I
MCF7	49	74	64.1	D
	56	61.6	53.5	D
	68	35.2	35.3	ND
MCFi	49	74.1	73.7	ND
	56	44.5	61.5	I
	68	27.1	40.7	I
MDA MB 231	49	101.2	80.2	D
	56	71.9	61.9	D
	68	48.9	28.8	D
MDA MB 231i	49	101.1	68	D
	56	102.5	54.1	D
	68	41.5	33.7	D

## Data Availability

The data presented in this study are held by the Department of Plastic and Reconstructive Surgery at Wake Forest School of Medicine but may be made available, subject to review, on request from the corresponding author.

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
