# Peer review of "The Impact of Silver Nanoparticle-Induced Photothermal Therapy and Its Augmentation of Hyperthermia on Breast Cancer Cells Harboring Intracellular Bacteria"

_pharmaceutics, 2023, doi:10.3390/pharmaceutics15102466_

Round 1

Reviewer 1 Report

The results obtained in this study are interesting and promising in the context of the application. The following questions should be addressed.

How common are intracellular bacteria within breast cancer cells?

The authors should reconsider the title of the work, which in my opinion is too general, and better specify the aim of this study.

In my opinion, the method for assessing the cytotoxicity of AgNPs to breast cell lines described by the authors is not reliable. To directly determine cytotoxicity, cell viability should be assessed using live/dead or resazurin staining. Additionally, on what basis did the authors choose the incubation time of AgNPs from breast cell lines (2 and 24 h)?

Different concentration of AgNPs was tested against breast cell lines after 2 h of exposure (10, 25, and 50 µg/mL) and after a 24 h of exposure (10, 25, and 50 µg/mL, 100 µg/mL, 250 µg/mL). Why?

In my opinion, the numbering of the figures is incorrect. Figure 5 should be at work earlier than figure 4.

Author Response

The results obtained in this study are interesting and promising in the context of the application. The following questions should be addressed.

We thank the reviewer for their thoughtful consideration of our work and the acknowledgement that the concepts show promise. We have addressed each of the reviewer’s comments below and changes in the text are indicated in red.

1.- How common are intracellular bacteria within breast cancer cells?

This is an excellent question. Although clinical breast cancer samples have identified that bacteria resides intracellularly, there is no statistical data currently available to determine how much bacteria in breast tumors is intra- or extra- cellular.  Breast cancer tissue harbors more bacteria than non-tumorigenic breast tissue, but the amount inside cells has yet to be quantified from clinical samples. We have included language in the discussion explaining that clinical data is not yet available to quantify the extent of intracellular burden. We have also included language that our results show that 1 out of every 10,000 MCF 10A cells is infected, and 1 out of every 100 tumorigenic cells is infected using P. aeruginosa as the infecting pathogen.  This may vary depending upon the bacterial strain and species, which the text further elaborates upon.

2.- The authors should reconsider the title of the work, which in my opinion is too general, and better specify the aim of this study.

Thank you for the suggestion that the title should be improved to be more specific for this work.  We have changed the title to ‘The Impact of Silver Nanoparticles-induced Photothermal Therapy and Augmentation of Hyperthermia on Infected Breast Cancer Cells and Intracellular Bacteria.’

3.- In my opinion, the method for assessing the cytotoxicity of AgNPs to breast cell lines described by the authors is not reliable. To directly determine cytotoxicity, cell viability should be assessed using live/dead or resazurin staining.

We thank the reviewer for their thoughtful consideration of cell viability measurements.  Trypsinization of the cells, and counting of viable cells is an old method that is rarely used due to its time-consuming nature. However, more current approaches such as crystal violet, MTS, MTT, CCK-8, live/dead and resazurin staining all stain both bacteria and the eukaryotic cells.  Therefore, using such stains overestimates eukaryotic cell numbers when cells have intracellular bacteria.  Therefore, these stains cannot be used to compare non-infected and infected cell viability, and direct counting is critical for accurate comparisons.  We have elaborated on this point in the text a bit more.

4.- Additionally, on what basis did the authors choose the incubation time of AgNPs from breast cell lines (2 and 24 h)?

Clinical hyperthermia treatments often use 30-120 min of elevated temperature.  Even though the PTT treatment is rapid, since there are multiple plates of cells, we sought to standardize the timing of the AgNP exposure to 2 h, since this is the maximum time that cells would be exposed while also being exposed to brief laser irradiation.  We then evaluated 24 h exposure of the cells to AgNPs since this is a more classical approach to determine cytotoxicity of a test agent. We have discussed the rationale for the timing in more depth in the text.

5.- Different concentration of AgNPs was tested against breast cell lines after 2 h of exposure (10, 25, and 50 µg/mL) and after a 24 h of exposure (10, 25, and 50 µg/mL, 100 µg/mL, 250 µg/mL). Why?

We already had the 24 h exposure data at all of these concentrations, and the toxicity at the high concentrations overwhelms the PTT effects, hence such high doses are not needed. We found the data to be valuable since there is resistance of the infected MDA MB 231 cells at 24 h.  Hence, for readers interested in using AgNPs against infected cells and not using PTT, we have included these results. In addition, the temperature elevations for 10, 25, and 50 ug/ml AgNPs are sufficient for a PTT effect.  Since only these concentrations will be needed and only need to be exposed for 2 h, during which time the cells will also be exposed to the laser, only these doses were evaluated at 2 h. We have revised the language of the text to better illustrate why different doses and timepoints for AgNP exposure where used.

6.-In my opinion, the numbering of the figures is incorrect. Figure 5 should be at work earlier than figure 4.

Thank you for this suggestion, we have reorganized the order of the figures to discuss the data from Figure 4 earlier

Reviewer 2 Report

Dear Editor and the Authors,

The findings of the paper can open up a new question how to design the nanoparticles for PTT to suppress cancer cell growth and/or eliminate cancer progression since AgNPs are thought to be toxic to the cancerous cells, but the findings revealed that the benefit has close relation with the cell line characteristics. Even though the paper was in a great-shape, there are only two things need to be discussed or at least clarified;

1- Hydrodynamic size and TEM-based size of AgNPs should not be quite similar, since DLS recognizes the organic load on the nanoparticles so they should look bigger. Also, larger nanoparticles give higher intensity in comparison to the smaller ones, therefore it is not easy to extrapolate the ratio of smaller/larger AgNPs based on the DLS result. What was the ratio of smaller AgNPs?

2- Chitosan is quite interesting molecule in biological applications, why wasn't there any discussion related to the presence of chitosan on the AgNPs?

Kind Regards,

Author Response

The findings of the paper can open up a new question how to design the nanoparticles for PTT to suppress cancer cell growth and/or eliminate cancer progression since AgNPs are thought to be toxic to the cancerous cells, but the findings revealed that the benefit has close relation with the cell line characteristics. Even though the paper was in a great-shape, there are only two things need to be discussed or at least clarified.

We thank the reviewer for their careful consideration of our work and we appreciate their comments and concerns. We have answered their queries below and made changes in the manuscript text in red to highlight changes.

1- Hydrodynamic size and TEM-based size of AgNPs should not be quite similar, since DLS recognizes the organic load on the nanoparticles so they should look bigger. Also, larger nanoparticles give higher intensity in comparison to the smaller ones, therefore it is not easy to extrapolate the ratio of smaller/larger AgNPs based on the DLS result. What was the ratio of smaller AgNPs?

The reviewer is correct that TM and DLS data may not correlate depending upon the thickness of the chitosan coating.  Under TEM the chitosan coating is visible under high magnification, not shown in the current image, as a fairly electron-translucent fibrous material.  We have observed that the chitosan coating of the nanoparticles is thin, and the thickness of the coating depends upon the shape of the nanoparticle and molecular weight of the chitosan.  From other images that we have taken the chitosan coating is thin, and hence correlates with the DLS data.  The DLS data was not intended to determine the ratio of smaller and larger particles, but this is a valuable point to consider.  When the AgNPs are produced they begin with the use of spherical seed particles, that are often about 10 nm in diameter. The DLS data indicates a peak near 10 nm, most likely from unreacted seed material that did not grow into plates.  The text discussed this previously, but we have elaborated on it a bit further to include that the purification of the AgNPs involves centrifugation and the small nanoparticles are not able to be removed except under ultracentrifugation.  They can be easily removed as they remain in the supernatant.  We have also included reference to our previous work which shown TEM that there are two populations of AgNPs, both spherical and triangular, which corroborates the DLS data in the current work.

2- Chitosan is quite interesting molecule in biological applications, why wasn't there any discussion related to the presence of chitosan on the AgNPs

We thank the reviewer for their thoughtful consideration of the presence of chitosan on AgNPs.  Chitosan has been used as a stabilizing agent for silver and other types of nanoparticles previously and there is a lot of literature available.  Since we have previously published on this composition of AgNPs we did not elaborate on it further initially.  However, since chitosan is also inherently antimicrobial, this needs to be discussed. We have revised the text to discuss this point, and the potential minor flaw that chitosan only nanoparticles were not also evaluated.